# Elevated Expression and Activation of GPR15 in Immune Cells in Graves’ Disease

**DOI:** 10.3390/biom12121899

**Published:** 2022-12-18

**Authors:** Jing Zhao, Xuerong Liu, Jianbin Xu, Yudie Fang, Peng Du, Chaoqun Gao, Tiantian Cai, Zhaohua Gu, Qiu Qin, Jin’an Zhang

**Affiliations:** 1Department of Endocrinology and Rheumatology, Shanghai University of Medicine & Health Sciences Affiliated Zhoupu Hospital, Shanghai 201508, China; 2Department of Endocrinology, Nanjing Medical University Affiliated Wuxi People’s Hospital, Wuxi 214000, China; 3Zhoupu Community Health Service Center of Pudong New Area, Shanghai 201508, China

**Keywords:** GPR15, GPR15L, IL4, IL21, Tfh, Graves’ disease

## Abstract

GPR15 plays an important role in lymphocyte homing and is a key immune molecule to maintain organ immune homeostasis. Yet, no study on the association between GPR15 and Graves’ disease (GD) is available. In this study, we systematically investigated the expression of GPR15 in different types of immune cells and different tissues of GD patients. We found that the expressions of GPR15 and GPR15L in peripheral blood of GD patients were increased compared with those in healthy controls. A flow cytometry analysis showed that GPR15 positive cells were mainly CD14+ monocytes and CD56+ natural killer cells (NK cells) of innate immunity, T helper cells and cytotoxic T cells of adaptive immunity. We also found that the expressions of GPR15 and GPR15L in the PBMC of GD patients were positively correlated with the Tfh-specific cytokines IL21 and IL4. In addition, immunohistochemistry showed that the level of GPR15 in thyroid tissue of GD patients was higher than that of the control group. Our results demonstrate for the first time that GPR15 is highly expressed in various immune cells in GD patients, suggesting that GPR15-GPR15L is associated with the activation and infiltration of proinflammatory immune cells in the thyroid tissue of GD patients.

## 1. Introduction

Graves’ disease (GD) is a syndrome of enlarged and hyperactive goiters (Graves’ hyperthyroidism), ocular abnormalities, and localized dermopathy lesions. As an organ-specific autoimmune disease governed by both genetic susceptibility and environmental factors, it affects 2% of women and 0.2% of men worldwide [1]. Thyrotropin receptor (TSHR) is a G-protein coupled receptor mainly expressed in thyroid follicular cells. In GD, the autoimmune tolerance mechanism is disrupted, and TSHR stimulates B cells to produce TSH receptor antibodies (TRAb) [2]. The thyroid tissue of patients with GD exhibits a typical lymphocytic infiltration, with the infiltrating lymphocytes releasing pro-inflammatory cytokines and activating TSHR-reactive immune cells [3,4]. The presence of TSHR antigen-reactive T cells and the production of TRAb are the hallmarks of GD [3,5,6]. However, the detailed pathogenesis of local thyroid infiltration by lymphocytes and the generation of thyroid auto-reactive lymphocytes are still obscure.

G protein-coupled receptor 15 (GPR15) is a guanine nucleotide-binding protein-coupled chemoreceptor (GPCR) initially identified for its similarity to the GPCR family [7], which is also recognized as the HIV and simian immunodeficiency virus co-receptor [8]. Recently, the chromosome 10 open reading frame 99 gene (C10orf99) was defined as the natural ligand for GPR15 and featured some similarities to the chemokine CC family, thus also known as GPR15L [9]. GPR15 acts as a lymphocyte transport receptor, mediating lymphocyte homecoming to tissues. Therefore, it is not difficult to understand the tissue specificity of GPR15L-GPR15 distribution under physiological conditions. That is, GPR15 is highly expressed in tissues in contact with the external environment, such as the skin and mucosal epithelium, to maintain the residence of immune cells and immune barrier function [10]. It is also easy to understand the overexpression of these molecules in skin inflammation and intestinal inflammation such as psoriasis, atopic dermatitis, and lichen planus [10,11]. The enrichment of GPR15+ T cells in the blood caused by smoking may indicate tobacco-induced chronic systemic inflammation [12,13]. In our previous animal experiment, by transcriptome sequencing, we accidentally found that GPR15 expression was increased in the thyroid tissue of Hashimoto’s thyroiditis model mice [14]. This is of great interest to us because the high expression of GPR15 might be associated with ectopic infiltration of lymphocytes in autoimmune thyroid diseases (AITD), while targeting this molecule may lead to the development of new therapeutic strategies for AITD.

## 2. Materials and Methods

### 2.1. Study Population

We recruited 56 patients with GD and 74 age- and sex-matched healthy controls in this study (IHC subjects were excluded). The diagnosis of GD should be made by a combination of clinical symptoms and laboratory findings [15]. Clinical symptoms included heat intolerance, hyperhidrosis, increased appetite, weight loss, muscle weakness, fatigue, tremor, and diffuse goiter. There were no clinical manifestations of pretibial myxoedema in the patients enrolled in this study, and none of the patients complained of itchy skin at visit. Laboratory test results included increased free thyroxine (FT4) and/or free triiodothyronine (FT3), decreased basic thyrotropin (TSH) levels, and positive TRAb. Exclusion criteria included age <18 years, smoking, current use of any medication that affects thyroid function, history of thyroid radioisotope therapy, concomitant cancer, infection, other acute or chronic medical conditions, and breastfeeding or pregnancy. All of the GD patients were recruited from the Department of Endocrinology, Shanghai Zhoupu Hospital. All healthy controls were collected from the Medical Examination Center of the same hospital. The clinical characteristics of the study subjects are provided in Table 1. The number of participates enrolled for each experiment differed, and details are shown in Appendix A. Our research was performed in conformity with the Declaration of Helsinki and was ethically authorized by the Ethics Committee of Zhoupu Hospital. Informed consent was obtained from all of the participants in this study. Serum levels of TSH, FT3, FT4, as well as TRAb and TPOAb levels were measured using an electrochemiluminescent immunoassay (Roche, Basel, Switzerland). All clinical laboratory tests were performed at the central laboratory of Zhoupu Hospital.

### 2.2. Peripheral Blood Mononuclear Cells Isolation

Venous blood from the participants was collected in anticoagulated tubes with EDTA. Peripheral blood mononuclear cells (PBMCs) were isolated by Ficoll density gradient centrifugation and separated according to the lymphocyte isolation medium manufacturing instructions (DAKEWE, Beijing, China). Isolated PBMCs were used to extract the total RNA or were cryopreserved in 90% fetal bovine serum/10% dimethyl sulfoxide.

### 2.3. RNA Extraction and Quantitative Real-Time Polymerase Chain Reaction (qRT-PCR)

Total RNA was extracted from PBMCs isolated from 2 mL of anticoagulated blood samples using Trizol reagent (TakaRa, Dalian, China), and cDNA was synthesized by 1 μg total RNA with a PrimeScript RT reagent kit (TaKaRa, Dalian, China). The expressions of target genes were detected by qRT-PCR, performed with SYBR Premix Ex TaqTM II (TaKaRa, Dalian, China) in ABI PRISM 7500, primers shown in Appendix A. The expressions of individual genes were normalized to β-actin control, and relative expression levels were calculated using the 2^−ΔΔCT^ method.

### 2.4. Flow Cytometry

Cryopreserved PBMCs (stored in liquid nitrogen) were rapidly thawed in a 37 °C water bath. Once washed with staining buffer (BD Pharmingen, San Diego, CA, USA), PBMCs were dyed in the dark with fluorochrome-conjugated antibodies against CD3, CD4, CD8, CD14, CD19, CD16/CD56, CD25, CD127, CD45RA, CCR7, and GPR15 (Appendix A). PBMCs were labeled according to the manufacturer’s instructions. Fixable Viability Dye eFluor 780 (eBioscience, San Diego, CA, USA) was used to assess the viability of PBMCs. Finally, cells were analyzed with Fortessa X20 after re-washing with wash buffer (BD Pharmingen, San Diego, CA, USA).

### 2.5. Enzyme-Linked Immunosorbent Assays (ELISA) for Plasma GPR15L

Plasma was separated from EDTA-anticoagulated whole blood by centrifugation. The whole blood samples were centrifuged at 300× *g* for 5 min at 4 °C, and the supernatants were then collected and continually centrifuged at 12,000× *g* for 5 min. After separation, plasma was immediately frozen at −80 °C until quantification. GPR15L in plasma was measured using a commercial sandwich ELISA kit (MEIMIAN, Yancheng, Jiangsu, China) according to the manufacturer’s instructions.

### 2.6. Immunohistochemistry (IHC) for GPR15

Immunohistochemistry was performed in five GD and six thyroid benign nodule samples from our institution. Briefly, thyroid tissue sections embedded in paraffin blocks were dewaxed and hydrated through an ethanol series, treated with 1 × citrate buffer for antigen recovery, blocked, and immunostained with primary antibody (anti-GPR15, rabbit, 1:100, Invitrogen, Carlsbad, CA, USA, PA5-32802) overnight at 4 °C. On the next day, the samples were stained with secondary antibody conjugated by horseradish peroxidase (Jackson ImmunoResearch Laboratories, West Grove, PA, USA, 1:10,000) and visualized with a DAB peroxidase substrate kit.

### 2.7. Statistical Analysis

Continuous variables with normal distribution were described as mean ± standard deviation (SD), and an independent samples t-test was performed to analyze between-group differences. Non-normal distribution data were reported as median (25th–75th percentile) and analyzed using a Mann-Whitney U-test. Data analyses were performed using R software (version 3.6.3), and R packages ggplot2 (version 3.3.3), with *p*-values < 0.05 being regarded as statistically significant differences. Flow cytometry experimental data were analyzed with FlowJo software (version 10.8).

## 3. Results

### 3.1. GPR15 and GPR15L Expression Levels Were Elevated in Patients with GD

During the initial phase of the study, we explored whether GPR15 is differentially expressed between the groups compared. We found that GPR15 mRNA expression in PBMC was not different in Hashimoto’s thyroiditis patients compared with healthy donors (*p* > 0.05, Appendix A), but increased in GD patients. To further investigate the effect of GPR15 on Graves’ disease, we expanded the sample size. The mRNA expressions of GPR15 and GPR15L increased significantly in the PBMC of the GD group compared to those in the health controls (both *p* < 0.001, Figure 1A). We also confirmed that GD patients had a significantly higher level of plasma GPR15L compared with healthy controls (*p* < 0.001, Figure 1B). To evaluate the GPR15+ cell differential distribution between GD patients and healthy controls, flow cytometry was performed on PBMCs in those two groups. Our results showed that GD patients possessed a higher frequency of GPR15+ cells in the peripheral blood (*p* < 0.001, Appendix A).

### 3.2. The Frequency of GPR15+ Cells Increased in Specific Subsets of PBMCs of GD Patients and the Expression of GPR15 Correlated Positively with IL4 and IL21

#### 3.2.1. The Frequency of GPR15+ Cells Increased in T cells, NK Cells, and Monocytes

To detect GPR15 expression in various immune cell groups, we applied multicolor flow cytometry in PBMCs derived from GD patients and healthy controls. We confirmed that GPR15 was expressed on B cells (CD3−, CD19+), T cells (CD3+, CD19−), natural killer (NK) cells (CD3−, CD16-56+), and monocytes (CD14+) in human PBMCs (Figure 1C). We compared the GPR15+ cell percentage in each cellular subpopulation between GD patients and healthy controls. We found that the frequency of GPR15+ cells was increased in T cells, NK cells (*p* < 0.01 for both subsets) and was significantly higher in monocytes (*p* < 0.001) for GD patients compared to healthy controls. Although this frequency decreased in B cells, the difference had no statistical significance (*p* > 0.05) (Figure 1D).

#### 3.2.2. GPR15+ Th Cells and GPR15+ Tc Cells Were Increased in GD Patients

T cells, as the major subset of PBMCs, are highly heterogeneous and can be classified into cytotoxic T cells (Tc), helper T cells (Th), and regulatory T cells (Treg) according to their functional properties. Therefore, we further analyzed the distribution of GPR15+ cells in Th cells (CD3+CD4+CD8−), Tc cells (CD3+CD4−CD8+) and Treg cells (CD3+CD4+CD127−CD25+) in GD patients and controls (Figure 2A). We also compared the percentage of GPR15+ cells in distinct T-cell subsets between GD patients and healthy controls. In fact, our statistical analysis showed increased GPR15+ cells both in Th cells and Tc cells (*p* < 0.05 for both subsets) for GD patients compared to healthy controls, but the proportion of GPR15 + Treg cells did not differ between the two groups (*p* > 0.05) (Figure 2B).

To determine whether GPR15 expression in PBMCs was independently associated with Graves’ disease, we conducted a multinomial logistic regression analysis while controlling for potential confounding factors (age and sex). The analysis demonstrated that the frequency of GPR15+ cells in PBMCs was independently positively associated with GD (OR = 1.248, *p <* 0.05). The analysis in immune cell subsets paralleled the above findings with some subtle differences. Despite the positive association of GD with GPR15+ cell frequencies in Th cells (OR = 1.491, *p* < 0.05), NK cells (OR = 8.066, *p <* 0.01) and monocytes (OR = 2.843, *p <* 0.001), GD was not associated with GPR15+ cell frequencies in Tc cells (OR = 1.556, *p >* 0.05), which may be due to the small number of cytotoxic T in PBMCs. A summary of the multivariable logistic regression analysis can be found in Appendix A.

#### 3.2.3. GPR15 Expression Was Enhanced in the TCM Subset of CD4+ T Cells and Both TCM and TEM Subsets of CD8+ T Cells in GD Patients

We then used CD45RA and CCR7 to subdivide Th cells and Tc cells into effector memory (TEM: CCR7−CD45RA−), central memory (TCM: CCR7+CD45RA−), terminal effectors (TEMRA: CCR7−CD45RA+), and naïve (TN: CCR7+CD45RA+) T cells (Figure 3A). As showcased in Figure 3, the proportion of different subtypes in TH cells was %TN > %TEM > %TCM > %TEMRA (Figure 3B), while that in TC cells was %TEMRA > %TN > %TEM > %TCM (Figure 3C). We subsequently assessed the mean fluorescence intensity (MFI) of GPR15 in each cell subpopulation. The results are presented in Figure 3D,E. In peripheral circulating lymphocytes, GPR15 expression was elevated on the TCM subset in CD4+ T cells (*p* < 0.01) as well as on the TCM subset (*p* < 0.01) and TEM subset (*p* < 0.05) in CD8+ T cells in GD patients compared to healthy controls.

#### 3.2.4. The Expression of GPR15 Correlated Positively with IL4 and IL21 in GD Patients

Th cells are also known for their high plasticity and ability to differentiate into different subsets, which secrete specific cytokines to modulate the immune reaction and play a vital role in the autoimmune response. To assess the effector functions of GPR15+ cells, we measured the cytokine mRNA expression levels (interferon-γ, IL-4, IL-10, IL-17A, IL-21, IL-22) in PBMCs of GD patients. In a Spearman correlation analysis, we observed that GPR15 was positively correlated with IL-4 (r = 0.71, *p* < 0.001) and IL-21 (r = 0.68, *p* < 0.001), but not with interferon-γ (IFN-γ), IL-17, IL-10, or IL-22 (Figure 2C).

### 3.3. GPR15 Level Was Raised in Thyroid Tissue of GD Patients

To investigate the discrepancy in GPR15 expression in the thyroid, immunohistochemistry was performed on six thyroid tissue samples from GD patients and six samples of normal thyroid tissues (paratumoral tissues from six benign thyroid adenomas). Clinical characteristics of relevant subjects are presented in Appendix A. Brown staining indicated positive immunohistochemical results for GPR15 protein, and the staining intensity of the GD samples was higher than that of the control samples (Figure 4A). The mean integral optical density (IOD) and the total stained area were selected as the parameters for image analysis. Compared with that of the controls, the expression of GPR15 in GD thyroid tissue was higher, and the difference between the two groups was statistically significant (*p* < 0.05, Figure 4B).

## 4. Discussion

As a lymphocyte trafficking receptor, GPR15 mediates the colon-specific transport of regulatory T cells and effector T cells in mice and humans, and is associated with immune homeostasis and intestinal mucosal inflammation [16,17]. Previous studies have shown that GPR15 is upregulated in immune cells in the blood and affects tissues in several autoimmune diseases. For example, GPR15 expression is elevated in the peripheral blood and synovial tissue of patients with rheumatoid arthritis [18]. As with other autoimmune diseases, the systematic activation of immune cells and local infiltration in target organs are key steps in the development of GD. Our study found that GPR15L-GPR15 was not only highly expressed in PBMC but also in lesioned tissues of patients with GD.

Immune cells can be divided into innate immune cells and adaptive immune cells. Both monocytes and NK cells constitute essential components of the innate immune system. Previous studies described an increase of monocytes in the peripheral blood of GD patients [19], while a decrease in the proportion of natural killer cells was also observed [20]. Interestingly, our results showed that the proportion of GPR15+ cells increased in both monocytes and natural killer cells of the peripheral blood in GD patients. Coincidentally, our unpublished studies and those of others have found a systemic imbalance of macrophage polarization and intrathyroid macrophage infiltration in patients with AITD [21]. Therefore, the increased expression of GPR15 in the peripheral blood represents the activation of monocytes, and it is hypothesized that GPR15 mediates the local infiltration of such cells in the thyroid gland during GD. Regarding NK cells, the majority of published studies generally agree that NK cells in GD patients lose their ability to prevent GD progression due to impaired function [20,22,23]. Our flow cytometry results showed that GPR15+ cells were significantly increased in circulating NK cells in GD patients, suggesting that GPR15 might be involved in the pathological progression of GD by regulating the function of innate immune cells.

Although both B cells and T cells mediate adaptive immune responses, our results showed that the frequency of GPR15+ cells was elevated only in T cells of GD patients. Upon further analysis, we found that GPR15+ T cells in GD patients expanded in the Th and TC subsets, but not in Treg subsets. Some independent studies have suggested that GPR15-expressing effector T cells (TE), rather than Treg cells, mediate local colonic inflammation [17,24], while another study showed increased GPR15 expression in peripheral blood Treg cells, but not in TE cells in patients with ulcerative colitis (UC) [25]. In contrast, GPR15 expression was significantly enhanced on colonic T cells from non-inflammatory biopsies of UC patients, but not in inflammatory biopsies [25]. Meanwhile, one study showed that GPR15 molecules guide Treg cells into the colon tumor tissue, where they modify the tumor immune microenvironment and promote intestinal tumorigenesis [26]. Il-17-producing GPR15+ T cells were significantly increased in the cerebrospinal fluid of patients with smoking-induced relapsing multiple sclerosis, suggesting a proinflammatory effect of GLP15+ T cells [27]. For further analysis, both Th and TC cells were divided into more detailed subgroups according to their activation status. We eventually observed the enhanced expression of GPR15 in TCM subsets of Th and TC cells and TEM subsets of TC. In addition, we also found that GPR15 mRNA expression levels were significantly positively correlated with IL-4 and IL-21 mRNA expression levels in the PBMC of GD patients. IL-4 is one of the most abundant cytokines secreted by Th2 cells and plays a crucial role in Th2 differentiation [28]. In one of our previous studies, interleukin-21 was shown to play a crucial part in the pathogenesis of GD and was associated with disease activity [29]. Furthermore, the circulating Tfh in this study was dominated by memory Tfh, which is consistent with our previous experimental results. IL-21 and IL-4, essential cytokines secreted by Tfh cells, enable B cells to initiate extrafollicular and germinal center antibody responses, which are critical for affinity maturation and maintenance of humoral memory [30]. Some studies have found that the increased Tfh cells in the peripheral circulation of GD patients decreased significantly after treatment [31]. Similarly, we also found a high expression of GLP15 in GD thyroid tissue. Combing these findings, we hypothesize that GPR15 is involved in the homing of circulating Tfh and Th2 cells to the thyroid gland, and promotes the maturation of specific B cells and the production of TRAb in the pathogenesis of GD. Combined with these findings, we hypothesized that GPR15 is involved in the homing of circulating Tfh and Th2 cells to the thyroid gland and promotes the formation of ectopic germinal centers, the maturation of specific B cells and the production of TRAb in the pathogenesis of GD.

## 5. Conclusions

In conclusion, our study revealed for the first time that the enhanced expression of GPR15 in GD patients was mainly observed in adaptive immune cells and antigen presenting cells, and mediated the homing of these cells, especially Tfh cells, to the thyroid gland. Blocking the GPR15-GPR15L effect is expected to be a potential treatment for GD.

## Figures and Tables

**Figure 1 biomolecules-12-01899-f001:**
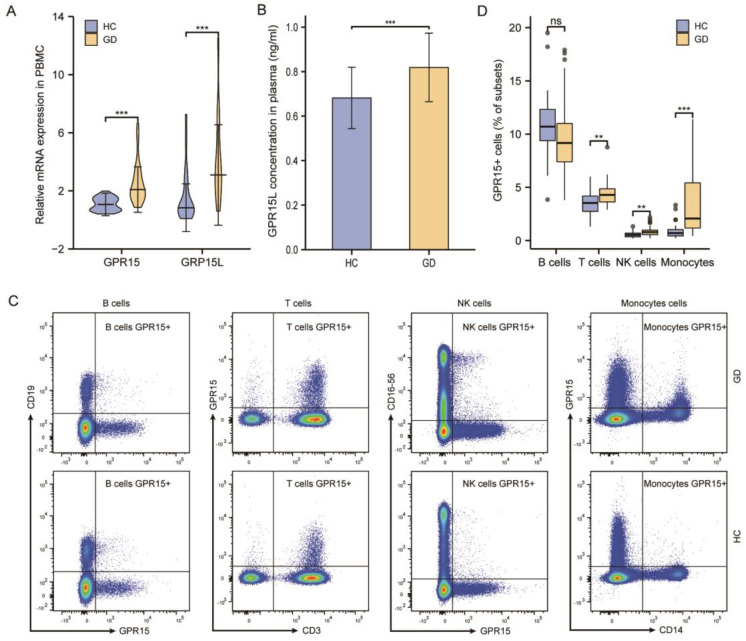
GPR15 and GPR15L expression levels are elevated in patients with GD. (**A**) Relative mRNA expression of GPR15 and GPR15L in PBMC. (**B**) GPR15L concentration in plasma. (**C**) Representative flow cytometric plots of GPR15 expression by B cells (CD19+), T cells (CD3+), NK cells (CD16-56+), and Monocytes (CD14+). (**D**) Percentage of GPR15+ of B cells (CD19+), T cells (CD3+), NK cells (CD16-56+), and Monocytes (CD14+). (ns, *p* ≥ 0.05; **, *p* < 0.01; ***, *p* < 0.001; HC: healthy controls, GD: Graves’ disease).

**Figure 2 biomolecules-12-01899-f002:**
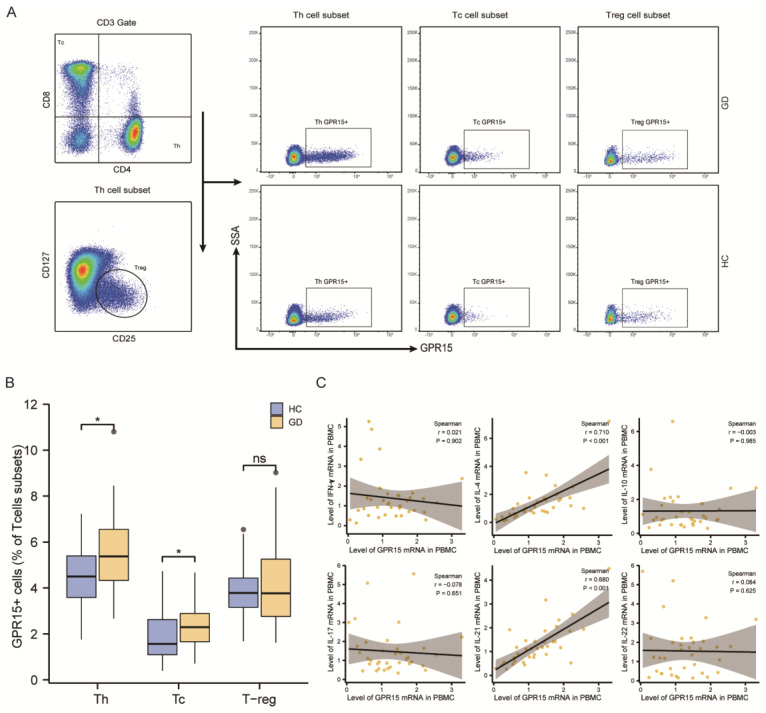
The frequency of GPR15+ cells increased on specific subsets of PBMCs from GD patients, and its expression positively correlated with IL-4 and IL-21. (**A**) Typical flow cytometry plots and gating for T cells subsets. (**B**) Statistical analysis of flow cytometry results in (**A**). (**C**) Correlations of GPR15 and cytokines expression in PBMCs from GD patients. (ns, *p* ≥ 0.05; *, *p* < 0.05; HC: healthy controls, GD: Graves’ disease).

**Figure 3 biomolecules-12-01899-f003:**
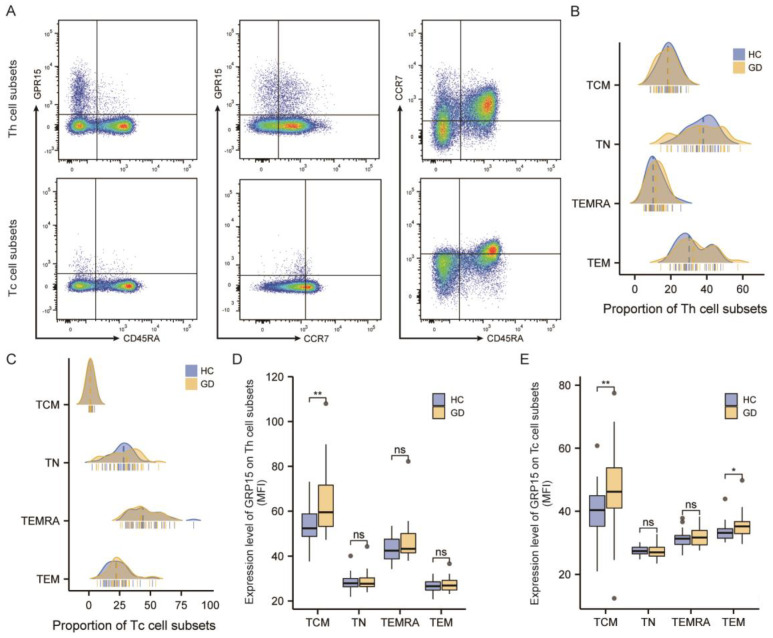
GPR15 expression was enhanced in several T-cell subsets of GD patients. (**A**) Typical flow cytometry plots and gating for subsets of T helper cells and cytotoxic T cells. (**B**) Proportions of T helper cells subsets in healthy controls and GD patients. (**C**) Proportions of cytotoxic T cells subsets in healthy controls and GD patients. (**D**) Statistical analysis of GPR15 MFI in subsets of T helper cells. (**E**) Statistical analysis of GPR15 MFI in subsets of cytotoxic T cells. (ns, *p* ≥ 0.05; *, *p* < 0.05; **, *p* < 0.01; HC: healthy controls, GD: Graves’ disease).

**Figure 4 biomolecules-12-01899-f004:**
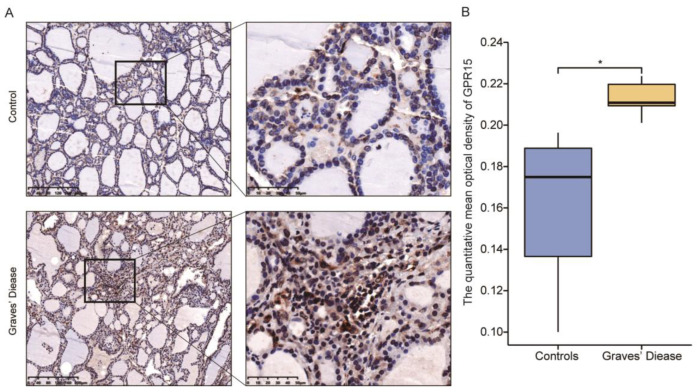
GPR15 levels were raised in thyroid tissue from GD. (**A**) Represents Immunohistochemical results of GPR15 of thyroid tissue specimens. (**B**) Statistical analysis of Immunohistochemical results. (GD = 5, Controls = 5). (*, *p* < 0.05).

**Table 1 biomolecules-12-01899-t001:** The clinical characteristics of the study subjects.

	Graves’ Disease	Health Control	*p* Value
Total	56	74	-
Age (years)	33.75 ± 11.00	35.10 ± 10.01	0.469
Gender (F/M)	45/11	48/26	0.052
FT3 (pmol/L)	21.29(12.16, 30.72)	-	-
FT4 (pmol/L)	34.52(27.75, 43.65)	-	-
TSH (IU/mL)	0.001(0.001, 0.003)	-	-
TRAb (IU/L)	8.59(4.75, 22.35)	-	-
TPOAb (IU/mL)	254.13(427.24, 62.59)	7.05(13.11, 5)	<0.001

F: female, M: male; TRAb: thyrotropin receptor antibodies.

## Data Availability

Not applicable.

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
