# Peer review of "Elevated Expression and Activation of GPR15 in Immune Cells in Graves’ Disease"

_biomolecules, 2022, doi:10.3390/biom12121899_

Round 1
Reviewer 1 Report
The authors of this article investigated the expression of G protein receptor15(GPR15) and of its natural ligand GPR15L in immune cells of patients with Graves'disease. They found an increased expression in peripheral blood and in thyroid tissue of these patients with respect to healthy controls. Moreover, flow cytometry analysis revealed that GPR15 were over-expressed in a variety of cells known to be able in triggering inflammatory/immune processes, being also positively correlate toTfh-specific cytokines of IL 21 and IL4. They conclude affirming that their results indicate that GPR15-GPR15L complex is associated with the activation and infiltration of proinflammatory immune cells in thyroid tissue of Graves' patients; thus blocking this effect could be a potential treatment of this disease.
COMMENT
GPR15 and its natural ligand GPR15L have been previously demonstrated in tissues of several organs, especially when affected by autoimmune/inflammatory diseases, but, their role in autoimmune thyroid diseases had not been explored to date. This is an interesting paper which contributes at clarifying the possible physiopathological role of GPR15-GPR15L in favoring the development of Graves' disease, However I have some concerns: - It should be better clarified whether the overexpression of GPR15-GPR15L complex at thyroid level may be specific to Graves diseases or it is also involved in the other autoimmune/inflammatory thyroid disesaes. To clarify these aspects it should be useful to investigate the expression of this complex in patients with Hashimoto thyroiditis or with subacute thyroiditis . - A recent paper demonstrated that GPR15L is a pruritogen factor ( Tseng and Horn. GPR15L is an epithelial inflammatio-derived pruritogen. Sci Adv 2022). It should be interesting to ascertain whether the patients enrolled in this study showed overexpression of GPR15L also in their localized dermopathy lesions and if they suffered from itching. - References 4 and 23 in Bibliography. It is not necessary to add to the title of the Journal(Thyroid), the sentence; official journal of the American Thyroid Association. Please delete.
.
Reviewer 2 Report
The manuscript about the elevated expression of GPR15 in immune cells in GD is interesting and offers some new insights towards the relation of GPR15 to GD.
However, I strongly suggest, that findings need to be confirmed within an independent set of samples before publication. This is even more important, because authors stated, that no other study has identified this type of association.
Besides that, I would like to recommend to adjust the statistical analysis. In addition to simple comparisons between GD patients and controls, authors should apply a linear model, that also considers co-variables such as age, gender and other parameters measure in GD patients and controls.
Minor points:
1. line 52: GPR15+T cells ? Shouldn't it be GPR15+ Tcells?
2. line 57: the term AITD has not been explained.
3. line 135: subsets - how did the authors define subsets or create subsets?
4. in general: the numbering/order of the figures is somehow confusing, i.e. Figure 1D is names before Figure 1C.
5. The method and detection of differences for flow cytometry experimental data have to be explained in more detail. It is not clear to the reader, how differences in Figures i.e. 1D and 2A have been detected and statistically evaluated.
Round 2
Reviewer 1 Report
I believe that the paper has been improved following the lines suggested by the reviewers and may now be reconsidered for publication on Biomolecules
Author Response
We would sincerely like to thank the reviewer for the encouraging comments on our manuscript.
Reviewer 2 Report
The figures have not reordered in a proper way. An it is still not clear, how different groups can be distinguished in the flow cytometric plots (i.e. Figure 1 C).
The statistical analysis has also not been changed. It is only a simple comparison of groups without multivariate model application.
